# Full-Scale Interface Friction Testing of Geotextile-Based Flood Defence Structures

**Emmett Klipalo** [1], **Mohsen Besharat** [2] **and Alban Kuriqi** [3,4,*]

1   School of Engineering, Arts, Science and Technology, University of Suffolk at Suffolk New College, Ipswich IP4 1LT, UK; emmett.klipalo@talktalk.net
2   School of Civil Engineering, University of Leeds, Leeds LS2 9JT, UK; m.besharat@leeds.ac.uk
3   CERIS, Instituto Superior Técnico, Universidade de Lisboa, 1049-001 Lisbon, Portugal
4   Civil Engineering Department, University for Business and Technology, 10020 Pristina, Kosovo
*   Correspondence: alban.kuriqi@tecnico.ulisboa.pt

**Abstract:** Open-topped woven polypropylene cellular containers filled with dense granular ballasts are often used as emergency flood defence structures. The effectiveness of these systems is highly dependent on the interaction with their bedding surface. The characteristics of the foundation will often govern the system's overall resistance to applied loading imposed by retained floodwater. However, the frictional relationship between polypropylene bulk bag flood defences and common bedding surfaces has not been extensively investigated. This study aims to reduce the reliance on arbitrary static friction coefficients by measuring and presenting actual data obtained through quantitative testing. This study presents the results of full-scale field testing to quantify the frictional resistance generated between filled polypropylene bulk bags and seven common bedding surfaces. Findings resulting from testing each interface scenario are expressed as coefficients of static friction. Test interfaces affording high frictional resistance comprised an unmade gravel road ($\mu = 0.74$) and grass ($\mu = 0.64$). Contrastingly, interfaces generating significantly lower frictional resistance were steel floated concrete ($\mu = 0.40$) and polypropylene plastic ($\mu = 0.40$). Test interfaces involving asphalt ($\mu = 0.54$) and tamped concrete ($\mu = 0.56$–$0.58$) were also investigated. This study recommends new friction coefficients necessary to characterise the structural stability analysis of bulk bag flood defences with greater accuracy. Practical advice based on experimental observation and field design experience is also given.

**Keywords:** climate change; flood protection; flood policy; hybrid infrastructure; urban floods





## 1. Introduction

Flood risk management authorities are under greater pressure as available resources are further stretched to keep pace with the continuous demand for flood protection [1–3]. In times of need, protecting residential and commercial communities from flooding relies on an extensive array of permanent and temporary defence infrastructure [4,5]. Ogunyoye and van Heereveld [6] present three main protection types:

- Permanent
- Temporary
- Demountable

Temporary Flood Protection systems form a key part of the United Kingdom's ability to combat flooding and are categorised as shown in Figure 1; a detailed illustration of these measures is given in Figures A1–A4.

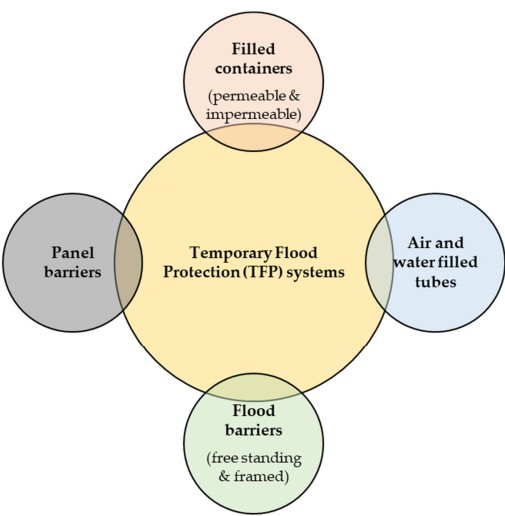

**Figure 1.** Classification of temporary flood protection systems, derived from Ogunyoye and van Hereveld [6].

Climate change is resulting in intensive and frequent events and transformations. Global warming is not only happening, there is scientific consensus that it is intensifying. As part of a concerning future climate, experts predict that extreme flooding will become more prevalent worldwide, with average sea levels projected to rise by over a meter by the end of the century [7–9]. Higher sea levels enable waves to carry greater energy to shore, which regularly exercises flood defences [10]. This further highlights the importance of a greater understanding of the structural stability of temporary flood defences as part of an adaptation strategy for future flooding events.

Alternative solutions must be sought in emergency scenarios where stocked temporary flood protection systems are inappropriate or cannot be sourced, transported, and erected in time. More traditional methods are adopted by flood risk management authorities, which often utilise the deployment of bulk bag structures, as illustrated in Figure 2 [11]. These structures consist of semi-permeable containers, often manufactured from polymer geotextiles/geosynthetics, including polypropylene filled with granular aggregates/ballasts [12]. Structural stability is obtained through their considerable self-weight and low center of gravity. Bulk bags have no fixed foundation; resistance to seepage relies principally on the fill material's density and the semi-permeable nature of the polypropylene container [13,14].

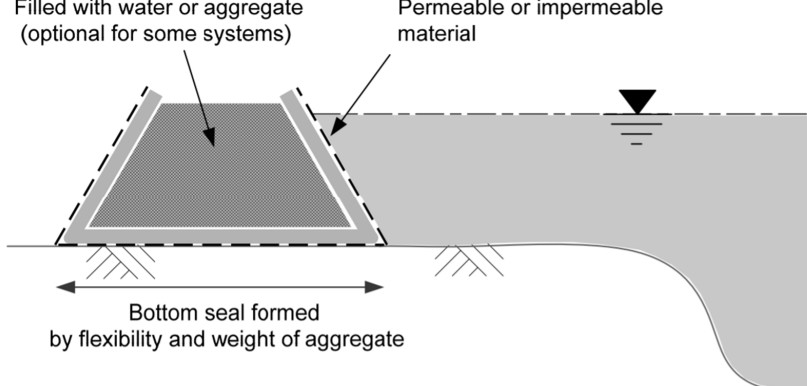

**Figure 2.** Permeable and impermeable bulk bag structures, derived from Ogunyoye et al. [15].

Flood risk management authorities favor the placement of bulk bags over traditional sandbagging techniques due to their reliability, versatility, and superior speed of construction [12,16]. Besides overturning and piping, sliding is a possible failure mechanism associated with small gravity structures such as bulk bags [17,18].

In 2009, HESCO Bastion units were deployed to combat flooding alongside the Red River in Fargo, North Dakota. HESCO Bastions are 'gabion-like' structural flood protection systems featuring linked non-woven polypropylene cellular baskets containing fill material [19]. While retaining floodwater, observers raised concerns that the stacked HESCO units were 'kicking out' at the base, leaning, and even sliding over the subsoil.

Ogunyoye et al. [18] suggest that sliding instability is the governing failure mechanism when bulk bag flood defences are built on firm impermeable subsoils. To obtain stability against sliding, bulk bag flood defences rely on resisting forces exceeding or equalising driving forces. Retarding frictional forces generated along the surface interface ($N\mu$) provides the predominant reaction in resistance to impending motion resulting from hydraulic force ($F$).

A review of the published literature reveals that previous studies designed to measure frictional resistance between geotextile temporary flood protection systems and various bedding surfaces are sparse [20]. Many sources advise using a single frictional coefficient ($\mu$) for all temporary flood defence stability calculations, irrespective of structure/surface type or condition [21,22]. Boon [12] recommends a 'shear coefficient' of 0.25 for use in stability calculations for all flood protection systems. Evaluations of various temporary flood protection systems performed by Duncan and Sleep [23] assumed a frictional coefficient of 0.45 for sliding stability calculations on all systems.

Krahn [24] devised a large-scale direct shear testing program to investigate six frictional interfaces within standard sandbag dykes. A 1 m² shear box powered by a hydraulic actuator was employed to apply a horizontal shearing force. Krahn [24] used woven slit film polypropylene (WSFPP) sandbags, a material used to manufacture bulk bags. Only a single bedding surface (i.e., grass sod) was tested. Krahn [24] reported that polyethylene sheeting on sod achieves an interface frictional coefficient of $\mu = 0.21$, and WSFPP on sod achieves $\mu = 0.42$. Offman and Blatz [25] extended Krahn's research by testing identical WSFPP polypropylene sandbags on a grass sod varying in temperature and moisture. Harms [26] reviewed the Syn-Tex Wave Breakers' resistance to sliding failure when hydrostatically loaded. Syn-Tex also manufacturers Super Sandbags', large geotextile sand-filled bags used for flood protection and individually placed with a rectangular cross-section. Friction coefficients between Wave Breakers and grass sod and sand surfaces are 0.50 and 0.46, respectively [26]. Moragues et al. [27] also undertook large-scale direct shear tests on six types of wave breakers. They found a significant relationship between the breaker types and flow characteristics and the wave energy dissipation on the slope.

The investigatory research presented within this paper was undertaken to quantify the sliding stability of polypropylene bulk bag flood defences founded on various bedding surfaces through the measurement of raw data to enable the calculation of static friction coefficients. This study aims to fill a perceived knowledge gap within the flood-fighting community. Flood defence professionals undertaking standard sliding stability assessments on bulk bag structures rely on arbitrary static friction coefficients assumed from generic material testing without known measured coefficients. This research ventured to reduce this reliance on arbitrary static friction coefficients through measurement and presentation of actual data.

A primary aim of this research was to extend the knowledge of static friction coefficients in the context of flood protection for use in the future planning, design, and deployment of temporary bulk bag flood defences. The authors' objective was to investigate a greater variety of surface interfaces than those analysed in previous studies. A full-scale field-testing program was devised to obtain the quantitative data required to satisfy the research aims.

## 2. Materials and Methods

The Environment Agency's Ely Operations depot was the chosen test facility. Bulk bag temporary flood protection systems are versatile and buildable across various sites upon variable ground conditions. Therefore, various bedding surfaces needed to be tested.



Test surfaces were selected based on their availability at the depot and the perception of where bulk bag temporary flood protection systems would likely be constructed within society. Table 1 displays the seven material interfaces that form each frictional test surface.

**Table 1.** Frictional surfaces tested within the fieldwork program.

| Frictional Surface 1 | Frictional Bedding Surface 2 | Ref. |
|---|---|---|
| Woven polypropylene plastic (Bulk bag material) | Dry interior concrete: Smooth (steel floated) | A |
| | Slightly damp exterior concrete: Tamped (wood floated)—parallel | B |
| | Slightly damp exterior concrete: Tamped (wood floated)—perpendicular | C |
| | Slightly damp macadam asphalt | D |
| | Mildly damp grass | E |
| | Damp unmade gravel road | F |
| | Dry polypropylene plastic | G |

A visual description of each surface is provided in Table 2. This research purposely focused on level surfaces only. Hence, test surfaces were selected with minimal perceptible inclination. In preparation for testing, a 2 m$^2$ frictional surface area was demarcated and swept clear of debris.

**Table 2.** Description of frictional surfaces tested within the fieldwork campaign.

| Ref. | Frictional Bedding Surface Description | Example Surface(s) |
|---|---|---|
| A | Smooth interior screed concrete finish with only rare visible delamination or imperfections. Surface asperities were of negligible-shallow depth ($\approx$0–3 mm). | Cast cill beams of floodgates. |
| B & C | A hardwearing exterior concrete feature surface tamping was designed to provide grip to those moving items around the depot yard. The occasional fine aggregate/pea shingle particle was noticeable within the troughed tamping. Asperities present in minor pitting were generally shallow-intermediate depth ($\approx$3–5 mm). | Floodgate's cill beams, coastal or riverside promenades, and footpaths. |
| D | A bituminous-based surface with a densely packed skeleton of rolled angular aggregate chippings. There were signs of surface course wear, thus exposing the chippings. Minor surface fracturing was visible. Asperities were generally of shallow depth ($\approx$0–3 mm). | Paved roads and paths within floodplains/next to watercourses and water bodies. |
| E | Grass of length generally <20 mm, with slightly patchy coverage and underlain with medium-firm topsoil. | Riverside grassed areas, flood banks, verges. |
| F | A rugged fine-particulate surface constructed from consolidated road planning. Some loose, mainly embedded sub-angular medium-coarse gravel was present. Asperities created from peaks and troughs of embedded gravel were generally of shallow-intermediate depth ($\approx$3–10 mm). | Unmade roads, tracks, and paths within floodplains/next to watercourses and water bodies. |
| G | Non-woven thin synthetic tarpaulin with no punctures or tears. | Damp-proof membranes. |

The testing used two woven polypropylene bulk bags (colored white and blue). The white bag was filled with road planning, and the blue bag with a 40 mm single-sized aggregate. The manufacturer's rated capacity for both bags was a 1000 kg safe working load (SWL). Like all objects under the gravitational influence, bulk bags have an associated maximum static friction force that varies depending on the bag's weight and the roughness of the interface between the bag and its bedding surface. Once the lateral driving force (i.e., hydraulic imposed by floodwater) exceeds the bulk bag's maximum static friction force, the surfaces slide relatively, and kinetic friction mobilises. The 'threshold of impending motion' was exceeded, as shown in Figure 3.

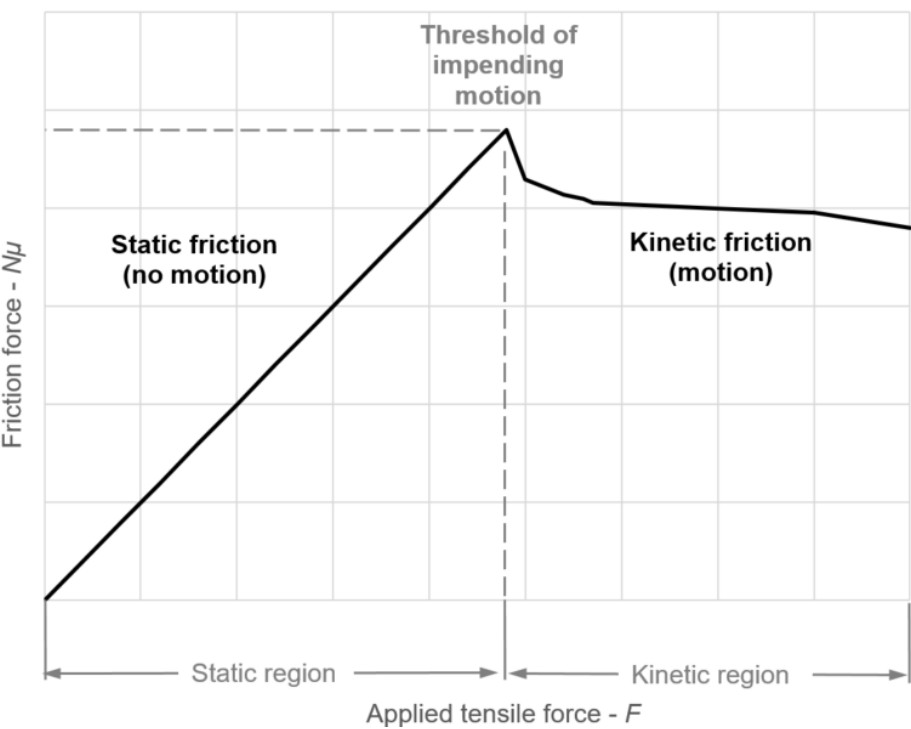

**Figure 3.** The indicative graphed output of a static friction test.

Within the discipline of statics, the coefficient of friction (μ) is a dimensionless ratio, given by Equation (1), of the force required to move an object from a state of rest (*F*) and its Normal force (*N*) [12,26].

$$\mu = \frac{F}{N} \tag{1}$$

When given two interfacing materials, the coefficient of friction refers to the shear strength of contacting junctions (i.e., interlocking asperities) and, therefore, its resistance to relative motion.

*Experimental Campaign, Procedure, and Apparatus*

Equation (1) requires the measurement of the two independent variables displayed in Figure 4. Full-scale field testing was favored instead of scaled laboratory due to the complexity of numerous influencing factors [12,28]. Soil conditions and foundation surfaces are not easily modeled in a laboratory environment. Conversely, full-scale field testing enabled the close representation of actual operational conditions and was favored [1,23]. The authors followed a procedure to collect the raw data by mimicking a hydrostatic scenario, which was the most effective method that could be achieved while working in conformance with existing resource and financial constraints. In this sense, the authors endeavored to implement measures to simulate a hydrostatic force as accurately as possible by including a backing board to distribute tensile (point) loading over the rear face of the bag, applying a slow, progressively increasing tensile force to the bag to simulate rising water. The digital load cell used during the field testing was calibrated independently. In reality, recording measurements by several repetitions showed that measurement errors have little to no effect on the final calculated static coefficient of friction.

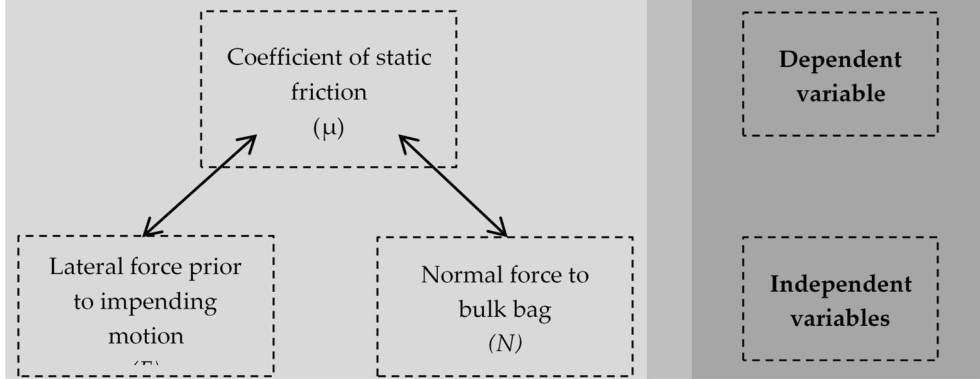

**Figure 4.** Inter-relationship between main dependent and independent variables.

Bulk bags may be stacked in formation to increase the width and height of a flood defence onsite. Therefore, initial testing on all surfaces (A–G) was conducted using a single bulk bag, followed by additional tests on selected surfaces (B, E, and G). Two bags are featured in a stacked formation. As a result, ten individual tests were conducted, as shown in Table 3. The authors believe that stacked testing on surfaces B, E, and G provides a good degree of variance in material interfaces. Therefore, comparative correlation to the unstacked test program can be drawn.

**Table 3.** Field testing matrix.

| Ref. | Frictional Bedding Surface 2 | Single Bag (Unstacked) | Two Bags (Stacked) |
|---|---|---|---|
| A | Dry interior concrete: Smooth (steel floated) | X | - |
| B | Slightly damp exterior concrete: Tamped (wood floated)—parallel | X | X |
| C | Slightly damp exterior concrete: Tamped (wood floated)—perpendicular | X | - |
| D | Slightly damp macadam asphalt | X | - |
| E | Mildly damp grass | X | X |
| F | Damp unmade gravel road | X | - |
| G | Dry polypropylene plastic | X | X |

Tests B and C investigated a tamped concrete surface where the direction of tamping was alternated between parallel and perpendicular. This was necessary to understand whether the tamping direction significantly impacted the surface's sliding resistance.

Bulk bag flood defences utilise polypropylene damp proof membranes (DPMs) beneath the bags to guard against piping and leakage. Test G was therefore specified to investigate the degree to which sliding resistance is affected when bulk bags are positioned on DPMs.

The field-testing procedure employed during this research aimed to simulate a hydro-static loading scenario where a smooth, progressively intensifying tensile force was applied to simulate rising floodwater (Figure 5). The hydrodynamic forces imposed by breaking waves and debris impact loading were not investigated during this testing program. Once filled, the weight of each bulk bag was recorded prior to testing to enable computation of the Normal force ($N$). The white and blue bags' filled unit weights were 656 kg (6.44 kN) and 684 kg (6.71 kN), respectively. As the angle of repose of each test surface was approximated to equal zero, a considered assumption was made whereby the Normal force ($N$) was taken to equal the bag's unit weight ($W$) directly.

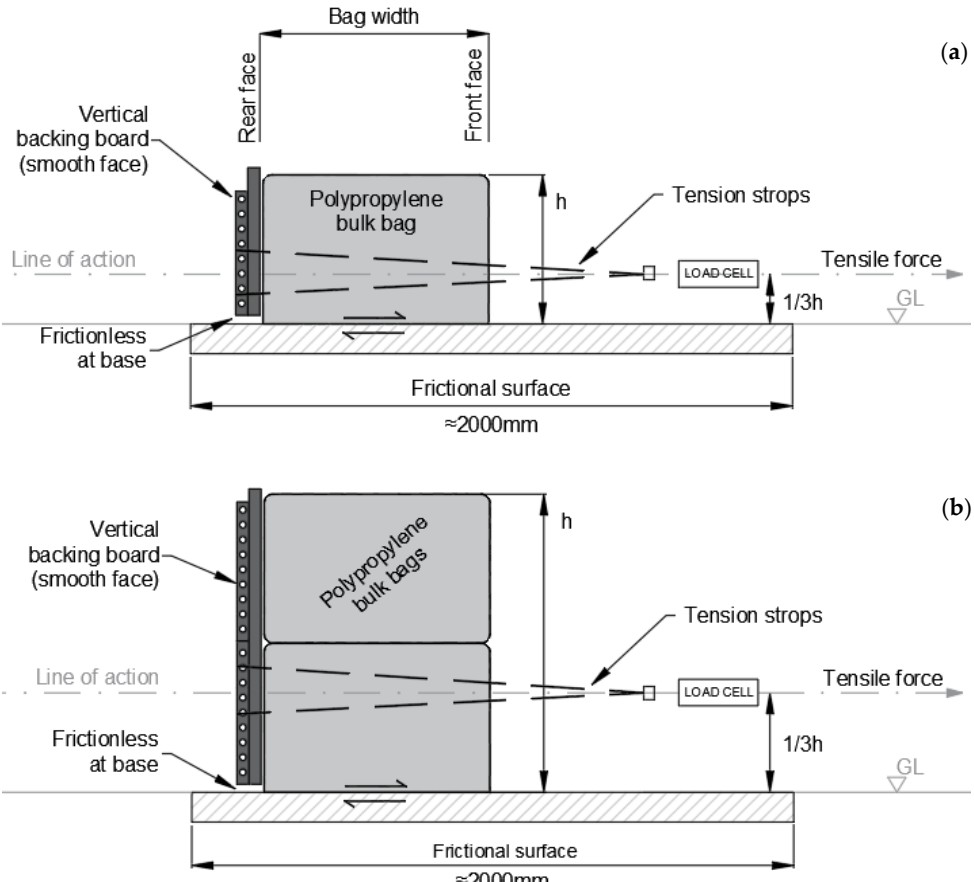

**Figure 5.** Illustration of field test set-up: (**a**) single bag unstacked; (**b**) double bag stacked.

Tension straps were slung around the bulk bag(s). A frictionless backing board uniformly distributed the applied tensile force over the entire rear face of the bag, thus replicating hydrostatic pressure imparted by retained floodwater [12,24,26]. The strops would cut into the bag without this backing board once tensioned. Therefore, an accurate representation of hydrostatic loading was unlikely to be achieved. The backing board was suspended above the foundational test surface to prevent the apparatus from generating additional frictional resistance. Where it was more awkward to suspend the backing board completely, a tarpaulin was laid upon the surface beneath to reduce the magnitude of any friction generated by the test apparatus.

Strops were parallel to the frictional surface at one-third of the imaginary water depth. For brevity during two-dimensional hydrostatic analysis, it can be considered that the force imparted by resting water on vertical surfaces such as seawalls or flood defences acts at the centroid of the triangular pressure diagram. A telehandler fitted with a 'hook-on carriage' attachment was employed to apply tensile force to the strops (Figure 6) until the threshold of impending motion was reached.

Although both exert erratic tensile forces and were not favored, the use of turf anchors and winches was considered. An impulsive lateral force was undesirable, as the threshold of impending motion may be missed or misrepresented. Instead, the telehandler's operator closely controlled the boom's position and rate of telescopic retraction.

The lateral force at the precise threshold of impending motion must be identified and measured to calculate the static coefficient of friction. A calibrated digital load indicator cell was connected to the telehandler and recorded the maximum tensile force within the strops prior to shear failure of the surface–structure interface [12,26]. Identifying the threshold of impending motion proved extremely challenging without this electronic function due to the load cell's sensitivity. The testing did not consider that harsh lubrication

of impermeable bedding surfaces would likely reduce static frictional resistance and the surface–structure interface.

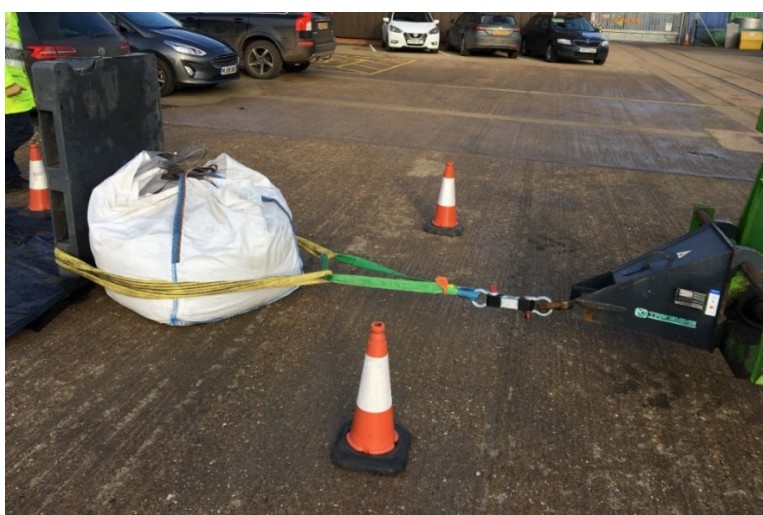

**Figure 6.** Single bulk bag subject to lateral tensile force during testing of surface C.

## 3. Results and Discussion

The tensile force was applied until the front face of the bulk bag(s) displaced up to 30 mm as the retarding friction force was overcome. Tests on each frictional surface were reset and repeated five times to reduce the chance of anomalous readings possibly resulting from stick–slip behavior.

Individual maximum load cell readings were first measured during an initial set of full-scale unstacked field testing. A single bulk bag was tensioned until displacement occurred [12]. These were followed by an additional set of testing, which investigated surfaces B, E, and G, whereby a second bulk bag was stacked above the existing bag. A summary of the raw data obtained during both testing sets is presented in Table 4. Mean deviations are presented for each surface's dataset to illustrate the dispersion of individual load cell readings around the mean.

**Table 4.** Summary of raw data—unstacked and stacked tests.

| Ref. | Frictional Bedding Surface 2 | No. of Readings Taken | Max Load Cell Reading Range (kN) | Mean Deviation (kN) |
|---|---|---|---|---|
| *Unstacked tests* | | | | |
| A | Dry interior concrete: Smooth (steel floated) | 5 | 2.53–2.63 | 0.026 |
| B | Slightly damp exterior concrete: Tamped (wood floated)—parallel | 5 | 3.45–3.71 | 0.072 |
| C | Slightly damp exterior concrete: Tamped (wood floated)—perpendicular | 5 | 3.57–3.67 | 0.028 |
| D | Slightly damp macadam asphalt | 5 | 3.26–3.61 | 0.098 |
| E | Mildly damp grass | 5 | 4.28–4.36 | 0.026 |
| F | Damp unmade gravel road | 5 | 4.59–4.94 | 0.090 |
| G | Dry polypropylene plastic | 5 | 2.69–2.84 | 0.050 |
| *Stacked tests* | | | | |
| B | Slightly damp exterior concrete: Tamped (wood floated)—parallel | 5 | 7.71–8.18 | 0.146 |
| E | Mildly damp grass | 5 | 7.75–8.61 | 0.247 |
| G | Dry polypropylene plastic | 5 | 5.02–5.40 | 0.092 |

A corresponding coefficient of static friction was calculated from each maximum load cell reading recorded, enabling a range of coefficients to be presented for each test surface in Table 5. Coefficients of friction are typically quoted between zero and one, where zero is non-frictional and one is highly frictional.

**Table 5.** Summary of calculated frictional coefficients—unstacked and stacked tests.

| Ref. | Frictional Bedding Surface 2 | Calculated Static Friction Coefficient Range (μ) |
|---|---|---|
| | *Unstacked tests* | |
| A | Dry interior concrete: Smooth (steel floated) | 0.39–0.41 |
| B | Slightly damp exterior concrete: Tamped (wood floated)—parallel | 0.54–0.58 |
| C | Slightly damp exterior concrete: Tamped (wood floated)—perpendicular | 0.55–0.57 |
| D | Slightly damp macadam asphalt | 0.51–0.56 |
| E | Mildly damp grass | 0.67–0.68 |
| F | Damp unmade gravel road | 0.71–0.77 |
| G | Dry polypropylene plastic | 0.40–0.42 |
| | *Stacked tests* | |
| B | Slightly damp exterior concrete: Tamped (wood floated)—parallel | 0.59–0.62 |
| E | Mildly damp grass | 0.59–0.65 |
| G | Dry polypropylene plastic | 0.38–0.41 |

Coefficients calculated from raw data obtained during the unstacked and stacked test programs are graphically displayed in Figure 7, along with standard deviation error bars.

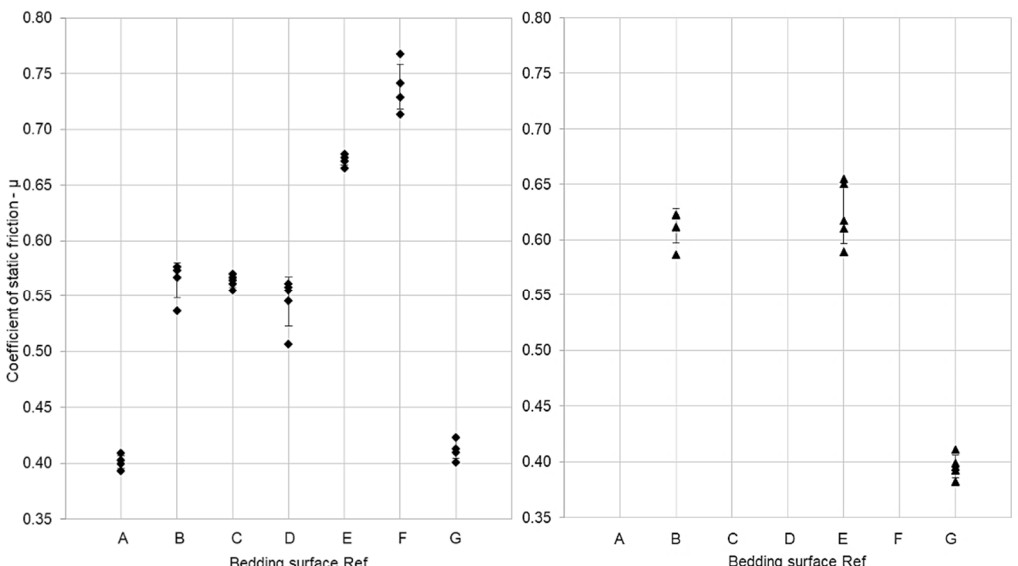

**Figure 7.** The standard deviation of calculated coefficients of static friction: unstacked tests (**left**); stacked tests (**right**).

A comparable friction coefficient range was achieved during the unstacked testing of the smooth concrete (ref A) and the polypropylene plastic (ref G). These surfaces had low asperity and the least frictional resistance [23]. Bulk bag flood defence structures founded upon these surfaces may require additional fixity depending on the intensity and longevity of loading [12]. The tamping direction in the exterior concrete surface has negligible influence on the bag's sliding resistance. When tested parallel and perpendicular to the tamping (ref B and C), a close-range of load cell readings was observed, and an

identical coefficient of friction was averaged. The depth of concrete tamping was very shallow and was thought to be the reasoning behind this.

Surface macrotexture directly influences the system–surface interaction, particularly the mechanical interlock that can be achieved between two surfaces [11,26]. The appreciable negative macrotexture of tamped concrete surfaces (refs B and C) contributed to a greater sliding resistance, especially when contrasted to surfaces with zero texture depth, such as the smooth concrete surface (ref A).

A high frictional resistance ($\mu = 0.67$) was measured during unstacked and stacked testing of grass (ref E). This may be attributed to slight undulations or irregularities in the underlying subsoil. Additionally, the bag's large concentrated weight likely compressed the subsoil beneath its footprint prior to testing, thus creating a 'dish' (Figure 8).

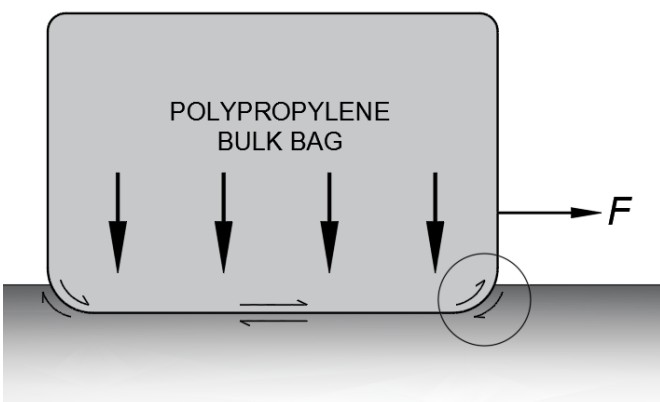

**Figure 8.** Exaggeration of proposed 'dish effect' encountered during testing of surface E.

Therefore, a greater tensile force was required to initiate the displacement of the bag. The highest static frictional coefficient was measured on the unmade gravel road (ref F). The unmade road featured greater texture depth than any other surface tested, produced by protruding medium-coarse embedded gravel. This texture had greater frictional interaction, as the bulk bag's fill material could interlock with the surface's embedded gravel [28].

The small friction coefficient measured from testing the polypropylene plastic surface (ref G) demonstrates that overlaying the natural bedding surface onsite during an emergency flooding scenario with a polypropylene DPM, as illustrated by Figure 9, is likely to reduce the frictional resistance generated along with the system–surface interface. Therefore, greater frictional resistance is likely to be achieved when the bulk bag directly interfaces with the site's natural bedding surface [24]. Surfaces B–F are common bedding surfaces found within society; each provided a greater sliding resistance than when the DPM was placed beneath the bag(s).

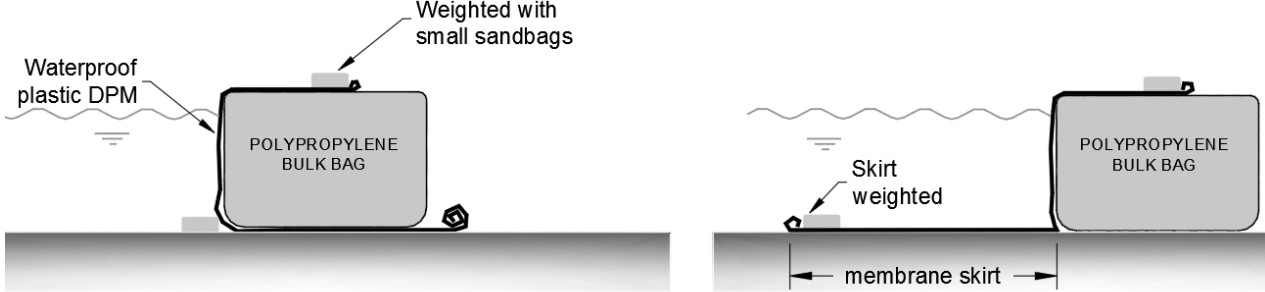

**Figure 9.** Cross-sectional illustration of the singular bulk bag with DPM.

We believe that the finite magnitude of the deviations displayed in Table 4 does not negatively impact the validity of the results [12,26].

Figure 10 shows that mean deviations from the stacked tests' raw data are larger on all surfaces than those calculated from the unstacked tests' raw data. The explanation for this could be apportioned to introducing a second bulk bag as an additional variable into the test procedure, which enhanced the potential for deviation [23,26].

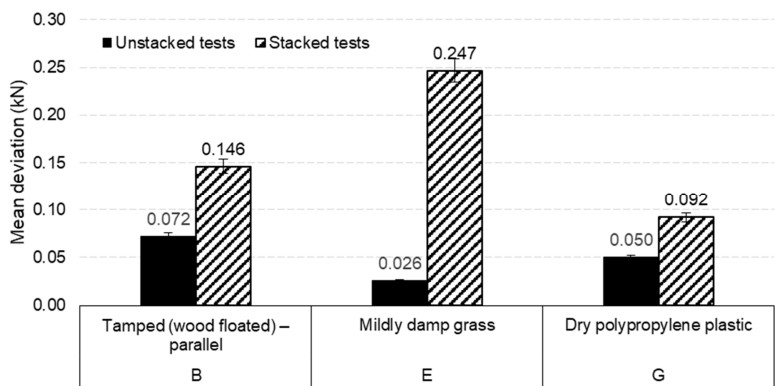

**Figure 10.** Mean deviation comparison—unstacked and stacked tests.

In order to compare static friction coefficients calculated from raw data from the stacked and unstacked test programs, an arithmetic mean was taken to produce a singular coefficient of friction. Table 6 displays these coefficients and a percentage variance between the values.

**Table 6.** Percentage variance between stacked and unstacked test data.

| Ref. | Calculated Static Friction Coefficient ($\mu$) | | Percentage Variance (%) |
| --- | --- | --- | --- |
| | Single Bag (Unstacked) | Two Bags (Stacked) | |
| B | 0.56 | 0.61 | 8.2 |
| E | 0.67 | 0.62 | 7.5 |
| G | 0.41 | 0.40 | 2.4 |

For coefficients obtained from testing of surface E, a greater frictional coefficient was calculated from the unstacked tests than the stacked tests. It is impossible to provide a definitive explanation for why this result occurred. However, an error was anticipated, as the system and apparatus employed during this study's field testing featured many sensitive, independent variables.

In summary, coefficients of friction obtained from the stacked and unstacked tests compare closely. This is a positive observation as, theoretically, increasing the weight of the bags (*W*) has no subsequent impact on the calculated coefficient of friction.

A sensitivity analysis was performed to amalgamate the static friction coefficients calculated from the stacked and unstacked field tests. This analysis culminated in Table 7, which presents the recommended friction coefficients for the seven bedding surfaces investigated.

**Table 7.** Summary of recommended coefficients of static friction ($\mu$).

| Ref. | Frictional Bedding Surface 2 | Friction Surface 1 |
| --- | --- | --- |
| | | Woven Polypropylene Plastic (Bulk Bag Material) |
| A | Dry interior concrete: Smooth (steel floated) | 0.40 |
| B | Slightly damp exterior concrete: Tamped (wood floated)—parallel | 0.58 |
| C | Slightly damp exterior concrete: Tamped (wood floated)—perpendicular | 0.56 |
| D | Slightly damp macadam asphalt | 0.54 |
| E | Mildly damp grass | 0.64 |
| F | Damp unmade gravel road | 0.74 |
| G | Dry polypropylene plastic | 0.40 |

## 4. Conclusions

Full-scale field testing investigated the friction between polypropylene cellular temporary flood defence structures and seven typical bedding surfaces. The presentation of static friction coefficients quantifies the frictional resistance generated at the system–surface interface during testing.

The research detailed within this paper focused solely on a hydrostatic loading scenario, simulating water at rest retained behind the temporary flood defence. In principle, hydrodynamic loading should be considered alongside hydrostatic loading when factoring in the stability of temporary flood defences. However, these bulk bag defences will likely be mostly constructed on sheltered rivers, lakes, reservoirs, and tidal estuaries, where hydrodynamic forces are not as prevalent. Floodwater in these environments will be mostly placid and will slowly rise to coincide with the peak of the surging tide or high water and then gradually recede. The effect of hydrodynamic loading on temporary bulk bag flood defence structures is the subject of the authors' future research.

Based on the fieldwork completed during this study, we present a diverse dataset of static friction coefficients, which are recommended as reference values for use in the analysis of temporary structures such as bulk bags. The test surfaces affording high frictional resistance were unmade gravel roads and grass. Contrastingly, the surfaces generating significantly lower frictional resistance were steel floated concrete and polypropylene plastic.

Bedding a temporary bulk bag flood defence structure on a plastic DPM will decrease the magnitude of friction generated along with the system–surface interface. With this arrangement, the structure's safety factor is inherently reduced; therefore, this should not be considered good practice.

Applying arbitrary static friction coefficients within temporary flood defence stability analysis without known measured coefficients remains ill-advised. Such coefficients are inaccurately based on generic material interface testing and may falsely misrepresent a surface's frictional resistance. The frictional coefficients presented within this study enable the design of bulk bag flood defences, which economise material requirements, as the system's frictional resistance can be estimated with greater certainty.

**Author Contributions:** Conceptualisation, E.K., M.B. and A.K.; methodology, E.K. and M.B.; software, E.K.; validation, E.K., M.B. and A.K.; formal analysis, E.K.; investigation, E.K., M.B. and A.K.; resources, M.B. and A.K.; data curation, E.K., M.B. and A.K.; writing—original draft preparation, E.K.; writing—review and editing, E.K., M.B. and A.K.; visualisation, E.K., M.B. and A.K.; supervision, M.B. and A.K.; project administration, M.B.; funding acquisition, M.B. and A.K. All authors have read and agreed to the published version of the manuscript.

**Funding:** This research received no external funding.

**Institutional Review Board Statement:** Not applicable.

**Informed Consent Statement:** Not applicable.

**Data Availability Statement:** No applicable.

**Acknowledgments:** The authors are sincerely grateful to the Environment Agency for the opportunity to undertake testing at their field operations depot and for the technical support afforded by its employees during the test campaign. Alban Kuriqi is grateful for the Foundation for Science and Technology's support through funding UIDB/04625/2020 from the research unit CERIS.

**Conflicts of Interest:** The authors declare no conflict of interest.

## Appendix A

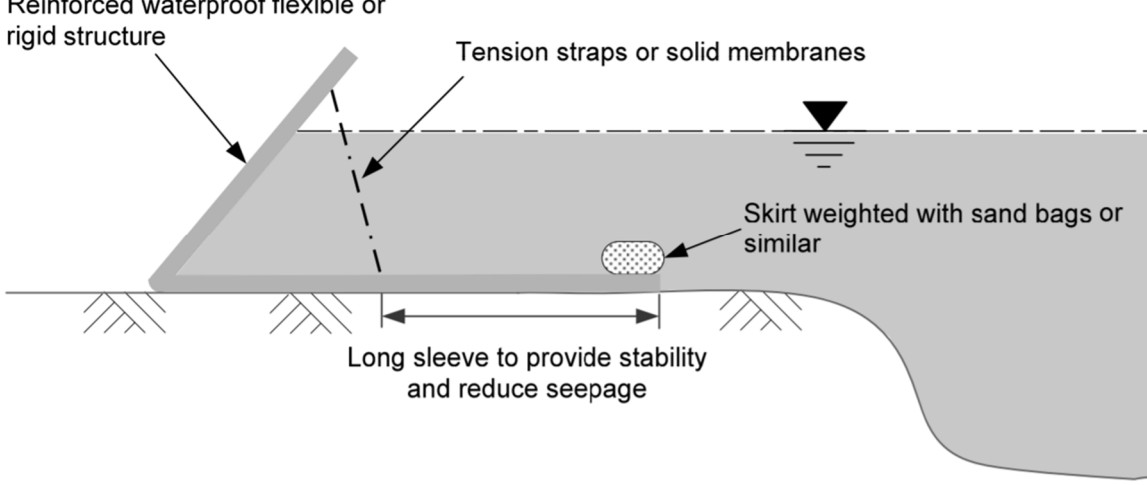

**Figure A1.** Air- or water-filled tubes, derived from Ogunyoye et al. [15].

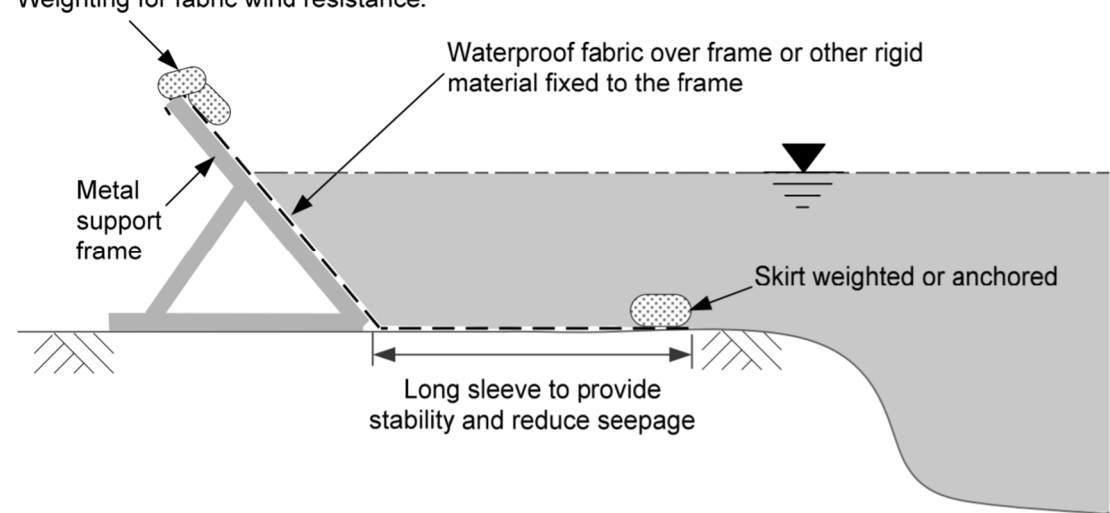

**Figure A2.** Free-standing barrier, derived from Ogunyoye et al. [15].

**Figure A3.** Frame barrier, derived from Ogunyoye et al. [19].

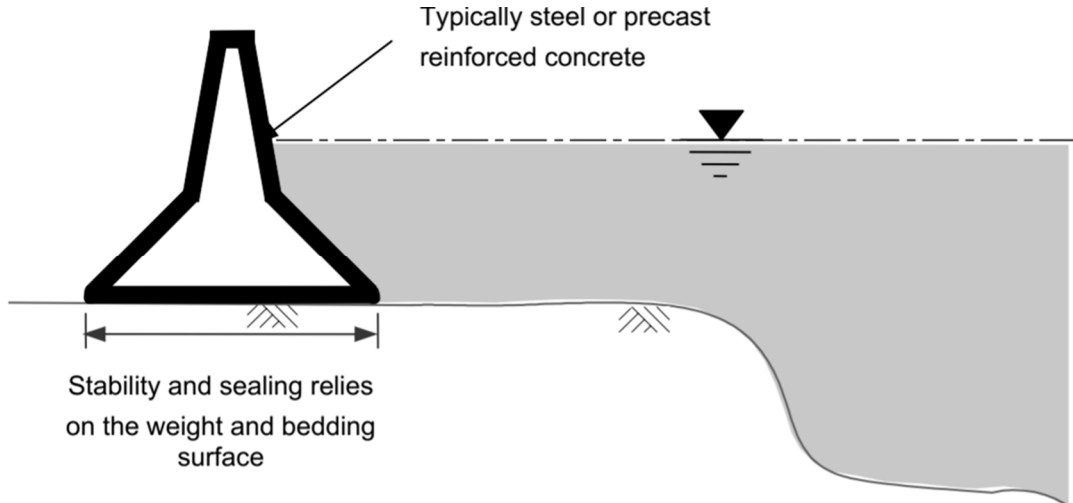

**Figure A4.** Panel barrier, derived from Ogunyoye and van Heereveld [6].

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
