# Peer review of "Full-Scale Interface Friction Testing of Geotextile-Based Flood Defence Structures"

_buildings, doi:10.3390/buildings12070990_

Round 1

Reviewer 1 Report

  1. The reviewer thinks the flood load can not be treated as a static load because the dynamic effects of moving water shall be considered. The ASCE 7-16 Chapter 5 gives some solutions for considering hydrodynamic loads, wave loads, and breaking wave loads. The reviewer is curious about how the authors consider this effect in your experiments or did the authors do some research about this? What is a possible solution when future researchers do the same experiments (such as introducing a dynamic pressure coefficient to the static load or decreasing the factor of safety calculated by the experimental results?) 
  2. Why did the authors choose these seven material interfaces? 
  3. Page 3, Tables 1 and 2, please re-edit them because some overlapped words are shown in this Table, and the line number should be on the right of each page. 
  4. Page 5, lines 161 to 163, please explain why only test surfaces B, E, and G were selected for the stacked tests.

Author Response

REVIEWER 1:

  1. The reviewer thinks the flood load can not be treated as a static load because the dynamic effects of moving water shall be considered. The ASCE 7-16 Chapter 5 gives some solutions for considering hydrodynamic loads, wave loads, and breaking wave loads. The reviewer is curious about how the authors consider this effect in your experiments or did the authors do some research about this? What is a possible solution when future researchers do the same experiments (such as introducing a dynamic pressure coefficient to the static load or decreasing the factor of safety calculated by the experimental results?)

In principle, hydrodynamic loading should be considered alongside hydrostatic loading when factoring in the stability of temporary flood defenses. However, these bulk bag defenses will likely be mostly constructed on sheltered rivers, lakes, reservoirs, and tidal estuaries where hydrodynamic forces are not as prevalent. Flood water in these environments will be mostly placid and slowly rise to coincide with the peak of the surging tide or high water and then gradually recede.

The authors’ performed a significant amount of desktop-based research into quasi-static and hydrodynamic loading characteristics and their impact upon temporary bulk bag flood defenses and vertical coastal/fluvial flood defenses in general and ended up with this conclusion.

The lead author’s research uncovered that a greater Factor of Safety should be achieved during the design of temporary bulk bag flood defenses, especially when these defenses are to be located in environments exposed to hydrodynamic conditions instead of hydrostatic environments due to the intensity and unpredictability of quasi-static and hydrodynamic loading.

As well as this, when calculating quasi-static and hydrodynamic forces to be imposed upon temporary bulk bag flood defenses during the design phase, empirical coefficients accounting for wave impact should be added to the formulae to account for dynamic effects. This is one of several ways that allowance for dynamic effects can be factored in during temporary flood defense stability assessments.

The authors would like to consider hydrodynamic effects such as drag forces, wave impact loading, and debris impact loading as part of a future research plan. Perhaps including the physical full-scale model tests of such effects if funding and resources are made accessible.

Part of this has been added to the manuscript at line 330.

  1. Why did the authors choose these seven material interfaces?

Test surfaces were selected based on possible surface materials and what has been suggested in relevant standards. That is coupled with the perception of where bulk bag temporary flood protection systems would likely be constructed within society (on roads, grass, ect).

  1. Page 3, Tables 1 and 2, please re-edit them because some overlapped words are shown in this Table, and the line number should be on the right of each page.

Thanks for highlighting this. The tables have been amended.

  1. Page 5, lines 161 to 163, please explain why only test surfaces B, E, and G were selected for the stacked tests.

An explanation was added to the manuscript on line 185.

Finally, we thank the esteemed reviewer for his excellent and valuable comments and questions. We hope that our answers will attract the opinion of the esteemed reviewer and that you and your journal will accept the article.

With best regards

Author Response

 REVIEWER 2:

The manuscript addressed a practical issue related to the structural stability analysis of flood defense structures. It presented good literature review and detailed description of the research methodology as well as the results of the full‐scale field tests. This is a good piece of work, and its results may be applied to develop and adopt a practical flood risk management plan. It is recommended to be published in this journal after minor revision following the suggestions listed below (all issues

must be properly addressed): 

Authors thanks the Reviewer 2 for kind words and encouragement.

  1. Section 2.1: The applicability of the experimental method used to determine the coefficient of friction depends on the accuracy of the instrument (digital load indicator cell) that has been used to measure the horizontal force. The authors did not verify the repeatability of the measurements using a statistical method. Is the measurement method the right procedure to mimic the hydrostatic load (evidence)? What is the overall accuracy of the instrument for measuring the resultant load due to the hydrostatic pressure? It is highly recommended to address this issue in order to improve the technical quality of the paper.

The digital load cell used during the field testing was calibrated independently.

The procedure used to collect the raw data by mimicking a hydrostatic scenario was the most effective method that could be achieved while working in conformance with resource and financial constraints.

The authors did not have access to full-scale hydraulic basins like the USACE use or hydraulic modeling software to simulate our loading scenario.

Section 2.1 explains our best endeavors to implement measures to simulate a hydrostatic force as accurately as possible (inclusion of a backing board to distribute tensile (point) loading over the rear face of the bag, applying a slow progressively-increasing tensile force to the bag to simulate rising water and applying this tensile force at 1/3 H).

In reality, recording measurements by several repetitions showed that measurement errors have little to no effect on the final calculated static coefficient of friction.

Appropriate explanations were added to section 2.1, line 170.

  1. Section 4: The authors determined the coefficients of static friction between filled polypropylene bulk bags and different types of bedding surfaces. In order to have a confidence in the results of the field tests, these coefficients must be compared with the results of previous studies for similar bedding surfaces. Also, a statistical method must be used to assess the deviation between their results and previous results reported in the literature.

The research was new; the surface materials had not been extensively researched before, so comparison with studies featuring an identical test method would be impossible based on the research conducted. 

However, the authors have already stated the coefficients measured by other creditable researchers such as Krahn, Boon, Harms, Offman & Blatz etc. in section 1 of the paper, so the reader can already compare the values.

Figure 10 and Table 6 compare coefficients obtained during this research’s stacked and unstacked tests in terms of mean deviation and percentage variance.

  1. Line 120: The author’s authors’ objective was to investigate a greater

Addressed as per attached final manuscript.

  1. Formatting or editing errors:
  • The authors must use the same citation style for EQUATION throughout the paper. They have to consult the author’s instruction of the journal.
  • Please correct the formatting and editing errors of the figure captions throughout the paper.
  • “Appendix” should be “Appendix A”

The above comments have been resolved. Thanks.

  1. Section 5: Conclusions: The study did not simulate a hydrodynamic loading scenario in the field tests. This limitation must be highlighted in Section 5.

This has been added to the manuscript on line 339. 

  1. Section: References: Please make sure that the titles of the journal articles and books (research report) and the names of the publisher are correctly written. For instance, see Ref# 26.

Harms, S. Evaluation of large sand‐filled geotextile containers as a temporary flood protection product. 2015. (is it a journal article or a book ???? name of the journal or the publisher???)

Addressed as per attached final manuscript.

Finally, we thank the esteemed reviewer for his excellent and valuable comments and questions. We hope that our answers will attract the opinion of the esteemed reviewer and that you and your journal will accept the article.

With best regards
